# Spatiotemporal Contrast Are Natural Urban Scene Learners

## Abstract

Street view imagery is a widely utilized representation of urban visual environments and supports various sustainable development tasks such as environmental perception and socio-economic assessment. However, it is challenging for existing image representations to specifically encode the dynamic urban environment (such as pedestrians, vehicles, and vegetation), the built environment (including buildings, roads, and urban infrastructure), and the environmental ambiance (such as the cultural and socioeconomic atmosphere) depicted in street view imagery to address downstream tasks related to the city. This work innovatively leverages temporal and spatial attributes of street view imagery to propose an unsupervised learning framework suitable for diverse downstream tasks. By employing street view images captured at the same location over time and spatially nearby views at the same time, we construct contrastive learning tasks designed to learn the temporal-invariant characteristics of the built environment and the spatial-invariant neighborhood ambiance. Our approach significantly outperforms traditional supervised and unsupervised methods in tasks such as visual place recognition, socioeconomic estimation, and human-environment perception. Moreover, we demonstrate the varying behaviors of image representations learned through different contrastive learning strategies across various downstream tasks. This study systematically discusses representation learning strategies for urban studies based on street view images, providing a benchmark that enhances the applicability of visual data in urban science.

## 1 Introduction

In recent years, unsupervised learning has demonstrated outstanding performance. By leveraging methods such as contrastive learning (He et al., 2020; Chen et al., 2020; 2021) and masked learning (He et al., 2022; Xie et al., 2022), it has achieved efficient image representation and exhibited excellence in classical computer vision tasks like image classification (Radford et al., 2021), object detection (He et al., 2022), and semantic segmentation (Wang et al., 2020a), surpassing the vast majority of supervised learning approaches. However, current unsupervised learning aims to encode as much semantic and structural information of objects and environments in a scene as possible (Park et al., 2023; Huang et al., 2024). This is not suitable for all downstream tasks in domains like street view-based urban environment understanding. For instance, in place recognition tasks (Lowry et al., 2015), the features are expected to focus only on place-invariant information, such as buildings and roads, filtering out dynamic information like lighting conditions, pedestrians, vehicles, and vegetation. In contrast, in tasks related to human perception of places (Dubey et al., 2016; Zhang et al., 2018), these dynamic elements are important. Moreover, tasks like socioeconomic prediction (Wang et al., 2020b) emphasize the spatially consistent expression of neighboring scenes.

In image representation learning, selectively encoding dynamic and static information in urban environments and the ambiance they create is highly important but inherently challenging (Cordts et al., 2016). Achieving precise encoding of such information typically requires separately labeling dynamic and static elements and using specific training strategies (e.g., masking dynamic elements when encoding static ones (Cheng et al., 2017; Wang et al., 2019)). However, both the labeling and training processes are fraught with difficulties. Factors such as lighting conditions, vegetation appearance, and ground litter are challenging to label objectively and consistently. This makes it nearly impossible to accurately represent these complex environmental factors using traditional datasets

(e.g., ImageNet (Deng et al., 2009), Places (Zhou et al., 2017)) and classical methods (supervised or unsupervised).

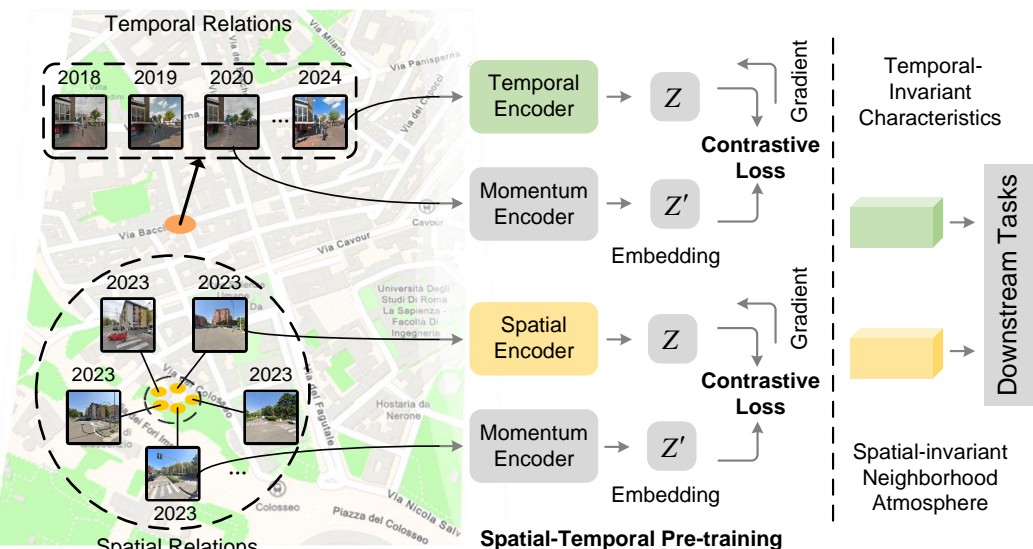

Figure 1: Spatial and temporal contrastive learning with street view images. Using street view images captured at the same location over time, contrastive learning tasks are designed to learn the temporal-invariant characteristics of the built environment; Using spatially proximate street view images from the same period, learning tasks are crafted to learn the spatial-invariant neighborhood ambiance, such as socioeconomic atmosphere.

Unlike existing large-scale datasets, street view imagery, as high-resolution urban visual dataset, possess unique spatiotemporal attributes that can capture both dynamic and static information in urban environments and the ambiance they form (Biljecki & Ito, 2021; Zhang et al., 2024). Therefore, in this work, we leverage these spatiotemporal attributes of street view imagery to propose a self-supervised urban visual representation framework based on street view imagery (see Figure 1). This framework aims to selectively extract and encode dynamic and static elements and their ambiance in urban environments according to the requirements of different downstream tasks, achieving precise representation of urban environments. Specifically, the framework is based on the following three hypotheses:

- **Temporal Invariance Representation**: At the same location, static elements such as buildings and streets do not change in images taken at different times, whereas dynamic elements like lighting conditions, pedestrians, vehicles, and vegetation present randomness in images taken at different times. Learning temporal invariant representations can retain the encoding of static elements while automatically filtering out information about dynamic elements. To capture this temporal invariance, we utilize the temporal attributes of street view imagery to construct positive sample pairs from historical street view images taken at different times at the same location. We expect that, after pre-training, the temporal encoder can learn stable features of the built environment. This method is suitable for tasks that rely on temporal stability, such as visual place recognition.

- **Spatial Invariance Representation**: At the same time, urban spaces at nearby locations usually exhibit similarity; the architectural styles and urban functions in adjacent areas are relatively consistent, while specific visual elements in images of nearby locations present randomness. Learning spatial invariant representations can encode the overall neighborhood ambiance within a specific spatial range while avoiding focus on any specific elements. To capture this spatial invariance, we leverage the spatial attributes of street view imagery to construct positive sample pairs from street view images taken in adjacent areas at the same time. We expect that, after pre-training, the spatial encoder can learn spatially

invariant neighborhood ambiance. This method is suitable for tasks that require spatial consistency, such as socioeconomic prediction.

- **Global Information Representation**: Besides temporal and spatial invariance, there are elements in urban environments that require holistic perception; these global features are vital for tasks involving human perception. To capture these characteristics, we construct positive sample pairs by applying data augmentation to the same street view image. We expect that, after pre-training, the model can retain the key elements of the scene and comprehensively capture the global information of the image.

We validate the effectiveness of these hypotheses across multiple urban downstream tasks. Experimental results demonstrate that different contrastive learning strategies can learn different types of features that are more suitable for their respective downstream tasks. We also conduct an in-depth analysis of the reasons behind the performance of different contrastive methods, further underscoring the importance of targeted learning strategies. This study systematically explores representation learning strategies in urban studies based on street view images, provides a valuable benchmark, and enhances the applicability of visual data in urban science.

## 2 RELATED WORK

### 2.1 SELF-SUPERVISED REPRESENTATION LEARNING

Self-supervised representation learning leverages the inherent structure within data to generate supervisory signals, thereby mitigating the need for extensive labeled datasets. A prominent approach in this field is contrastive learning, which has demonstrated significant success in learning robust representations. Methods such as InstDis (Wu et al., 2018), SimCLR (Chen et al., 2020), and the MoCo series (He et al., 2020; Chen et al., 2021) focus on contrasting positive pairs of similar instances against negative pairs of dissimilar instances to learn effective feature embeddings. In contrast, BYOL (Grill et al., 2020), SimSiam (Chen & He, 2021), and DINO (Caron et al., 2021) improve performance by avoiding negative samples altogether and adopting a self-distillation approach. These methods have achieved notable results in various visual tasks, such as image classification and object detection, showcasing the ability of self-supervised learning to perform exceptionally well with large-scale unlabeled data. However, despite these successes, existing self-supervised learning methods predominantly focus on static images without considering the spatiotemporal context inherent in certain datasets, such as urban environments captured over time and space. The lack of integration of spatiotemporal information limits the models' ability to capture dynamics over time and across spatial regions, especially in tasks requiring an understanding of both spatial and temporal dependencies. Therefore, there is a need for self-supervised learning approaches that effectively incorporate spatiotemporal information to enhance performance in such tasks.

### 2.2 SPATIOTEMPORAL CONTRASTIVE LEARNING IN VISION TASKS

Spatiotemporal contrastive learning enhances traditional contrastive learning by integrating both spatial and temporal information, enabling models to capture underlying relationships in unlabeled data that vary over space and time. Temporal contrastive learning excels in sequential data by differentiating between related and unrelated frames. For example, Contrastive Predictive Coding (CPC) (van den Oord et al., 2019) applies temporal contrastive learning by using consecutive video frames as positive pairs and shuffled or temporally distant frames as negative pairs, helping models learn temporal coherence. SeCo Manas et al. (2021) uses multi-season remote sensing images for self-supervised pre-training, enhancing model performance in remote sensing tasks. Spatial contrastive learning improves a model's ability to represent spatial scenes from various angles, perspectives, and locations. Multi-view contrastive learning approach is typically applied within a single scene from multiple angles at one location (Tian et al., 2020). Building on these concepts, geospatial contrastive learning contrasts data from different geographic locations or regions. By ensuring that data from similar spatial locations are closer in the feature space while data from different regions are more distant, models can more effectively capture spatial patterns and geographic features (Deuser et al., 2023; Ayush et al., 2021; Klemmer et al., 2024; Mai et al., 2023). This approach enhances the understanding of spatial relationships across wider geographic contexts.

## 2.3 STREET VIEW REPRESENTATION LEARNING FOR DOWNSTREAM TASKS

Street view imagery has been widely used in various urban downstream tasks, such as road defect detection (Chacra & Zelek, 2018), urban function recognition (Huang et al., 2023), and socioeconomic prediction (Fan et al., 2023). However, existing research on street view representation often relies on supervised models trained on datasets like Places365 (Zhou et al., 2017) or directly uses the pixel proportions of semantic segmentation results. These approaches fail to fully capture the rich semantic information embedded in street view imagery. Unlike natural images, street view imagery not only contains complex visual semantics but also encodes valuable spatiotemporal information in its metadata. Effectively representing this dual semantic nature—both visual and spatiotemporal—remains a significant challenge for improving its use in urban downstream tasks. Although a few studies have explored spatiotemporal self-supervised learning approaches to represent street view imagery (Stalder et al., 2024), these methods still have limitations. For instance, Urban2Vec (Wang et al., 2020b) incorporates spatial information into self-supervised training by constructing positive sample pairs based on nearest neighbors, while KnowCL (Liu et al., 2023) integrates knowledge graphs with contrastive learning to align locale and visual semantics, improving the accuracy of socioeconomic prediction using street view imagery. However, these approaches fail to explore the natural meanings of the spatiotemporal attributes of street view imagery and how to leverage these attributes to construct self-supervised methods suitable for various downstream tasks.

## 3 METHOD

The real world undergoes continuous changes across both temporal and spatial dimensions, yet these changes exhibit a certain level of continuity. In the temporal dimension, it is important to capture the invariant characteristics of a location as they evolve over time. Meanwhile, in the spatial dimension, the focus is on maintaining the consistency of the overall atmosphere within a specific spatial range. These temporal and spatial invariances are crucial for enhancing performance in various downstream tasks. In this section, we introduce the proposed spatiotemporal contrastive learning framework in detail (Figure 1).

Contrastive learning aims to learn feature representations from unlabeled data by contrasting positive and negative samples. The primary goal is to minimize the distance between positive samples and maximize the distance between negative samples within the feature space. Positive pairs are constructed by applying data augmentations to street view images. By optimizing the InfoNCE loss function, the model learns to reduce the distance between positive pairs in the feature space and increase the distance from negative samples, thus improving the feature representation learning. Given a query representation $q$ and a set of positive and negative keys $(k^+, k^-)$, the InfoNCE (van den Oord et al., 2019) loss is defined as:

$$\mathcal{L}_q = -\log \frac{\exp\left(q \cdot k^+/\tau\right)}{\exp\left(q \cdot k^+/\tau\right) + \sum_{k^-} \exp\left(q \cdot k^-/\tau\right)} \tag{1}$$

Here, $q$ is the feature representation of the query, $k^+$ is the feature representation of the positive sample, and $k^-$ is the feature representation of negative samples. The temperature parameter $\tau$ controls the scaling of similarities. The goal is to maximize the similarity between the query and the positive key $q \cdot k^+$ while minimizing the similarity between the query and the negative keys $q \cdot k^-$. Building on this contrastive learning framework, we introduce temporal and spatial contrasts for constructing positive pairs from street view images.

**Temporal Contrast.** Street view images captured at the same location but at different times differ from video frames because the intervals between shots are not fixed. Unlike remote sensing images, street view images taken at different times are not perfectly aligned in terms of position. Due to the typical spatial and angular shifts between images captured at different times, we impose restrictions on the conditions for positive temporal pairs: they must be taken within 5 meters of each other and have the same shooting angle. The historical street view image set for each location can be represented as $T = [t_1, t_2, \ldots, t_n]$, where $t_i$ denotes the images captured at different times. Since the number of images varies for each location, the value of $n$ differs accordingly. The aim of temporal contrast is to capture the invariant features of the same location over time. This means that even though the images are taken at different times, the model should learn to recognize the

consistent characteristics of the scene. To achieve this, we define positive pairs $(t_i, t_j)$, which are images that satisfy the aforementioned temporal conditions.

**Spatial Contrast.** Capturing the spatial consistency of an urban area is essential for accurately representing urban physical environment. Spatial consistency refers to the ability to recognize that different locations within the same urban area still represent the same underlying physical characteristics. To achieve this, we treat all street view images captured within a specific urban area as representing the same environment, even if these images are taken from different angles or slightly different positions. This approach allows the model to account for variations in location while preserving the overall atmosphere of the area. The set of street view images for a given urban area can be denoted as $S = \{s_1, s_2, \ldots, s_n\}$, where each $s_i$ represents an image captured within the defined area. These images collectively provide a comprehensive spatial representation of the urban environment. We randomly select two samples $(s_i, s_j)$ from the set $S$ and treat them as positive pairs. This encourages the model to learn that despite slight variations in shooting angle or position, the images are part of the same spatial context. By doing so, we enable the model to learn consistent and representative spatial features across the entire urban area.

## 4 EXPERIMENTS AND RESULTS

To validate our hypothesis, we first pre-train the models using datasets specifically designed for self-supervised contrastive learning, spatial contrastive learning, and temporal contrastive learning, respectively. We then evaluate the models on three distinct downstream tasks that reflect the characteristics of these contrastive learning models: visual place recognition (VPR), socioeconomic indicator prediction, and safety perception. Additionally, we conduct interpretability analyses on the features learned by the different contrastive models to gain a deeper understanding of the information the models focus on and how this impacts performance on urban downstream tasks.

### 4.1 SPATIOTEMPORAL PRE-TRAINING

Since the VPR and safety perception datasets include a wide range of street view samples from different cities, while the socioeconomic prediction task focuses more on local city characteristics, we constructed two separate datasets — a global version and a local version — for testing on different downstream tasks.

For the global version, to capture a broad spectrum of urban environments, we trained our self-supervised models on data collected from ten diverse and representative global cities. These cities were carefully selected to encompass a variety of geographical locations, cultural backgrounds, and urban forms, ensuring the diversity and richness of our training dataset. We historical images of ten global cities from the Google Street View (GSV) API which resulted in a total of over 42 million street view images used for pre-training. Detailed information about the data collection process and the composition of the dataset can be found in Sections A.1 and Section A.2.

For the local version, we selected street view data from Los Angeles to construct different contrastive datasets tailored to the specific needs of the socioeconomic prediction task in that city. The construction methods of datasets are similar to the global version.

Based on the street view pre-training datasets, we constructed three distinct contrastive datasets corresponding to different contrastive learning models for both global and local version: self-contrastive, temporal contrastive, and spatial contrastive datasets. To benchmark against the MoCo v3 baseline trained on ImageNet, each dataset was standardized to consist of 1 million image pairs. This uniform dataset size facilitates a fair comparison among the models by ensuring that each receives an equal amount of training data.

**Self-contrastive Dataset**. For the self-contrastive dataset, we randomly selected 100,000 images from each of the 10 cities, resulting in a total of 1 million images. Positive pairs were generated during training by applying data augmentation techniques to these images, following the settings used in MoCo v3 (Chen et al., 2021). Additionally, for the local version, we constructed a self-contrastive dataset based solely on Los Angeles using the same method.

**Temporal Contrastive Dataset.** In constructing the temporal contrastive dataset, we randomly selected 100,000 street view sampling points from each of the 10 cities, totaling 1 million sampling

points. At each sampling point, we retrieved images taken at different times but from the same shooting angle. Two images were randomly selected from the temporal sequence to form a positive pair, resulting in 1 million temporal positive pairs. Similarly to the self-contrastive dataset, we constructed an additional temporal contrastive dataset based solely on Los Angeles using the same method.

**Spatial Contrastive Dataset.** For the global spatial contrastive dataset, we defined a 100-meter buffer zone as a unified urban area. From each buffer zone, we randomly selected two images to form positive pairs. Out of all the spatial positive pairs generated, we then randomly selected 1 million pairs to create the spatial contrastive dataset. It is important to note that we did not impose any restrictions on the shooting angle for positive pairs, allowing the model to focus more on the overall ambiance of the urban environment rather than specific street layouts. Similarly, for the local version, since the socioeconomic dataset is based on block groups, we defined each block group as an urban area and constructed positive pairs based on the block group boundaries.

**Training.** We use AdamW (Loshchilov & Hutter, 2019) as the optimizer, a common choice for training ViT base (Dosovitskiy et al., 2021) models, with a weight decay of 1e-6. For each dataset, we use a mini-batch size of 1024 and an initial learning rate of 6e-6. The model is trained for 300 epochs, starting with a 40 epoch warmup(Goyal et al., 2018), followed by a cosine decay schedule for learning rate decay (Loshchilov & Hutter, 2017). Training the ViT Base model for 300 epochs on 4 Nvidia A800 GPUs takes approximately 71 hours.

## 4.2 VISUAL PLACE RECOGNITION

VPR is a crucial urban task that aims to identify specific locations based on visual input. This task requires the removal of temporal disturbances to focus on stable information that does not change over time, demanding feature extraction that effectively distinguishes constant characteristics in the environment to improve recognition accuracy.

To evaluate the model's performance in VPR tasks, we used several benchmark datasets: CrossSeason (Mans Larsson et al., 2019), Essex (Zaffar et al., 2021), Pitts250k, Pitts30k (Arandjelović et al., 2018), SPED (Chen et al., 2018), and MapillarySLS (Warburg et al., 2020) datasets. The model was tested by freezing the backbone of the pre-trained ViT and extracting the [CLS] token for VPR tasks. We assessed performance using the Recall@K metric, measuring the model's ability to correctly identify query image locations among the top-k most similar database images.

The GSV-Temporal model demonstrates exceptional performance on the CrossSeason dataset, achieving a recall value of 100% across all K values. This indicates its robust capability in cross-season VPR tasks. In contrast, GSV-Self and ImageNet-Self exhibit significantly lower performance, suggesting their inability to effectively capture temporal features. On the Essex dataset, GSV-Temporal maintains a recall value exceeding 75%, with values of 99.05% for both K=20 and K=25. This highlights its sensitivity to dynamic changes in the environment, outperforming other models in this context. In the Pitts250k dataset, GSV-Temporal consistently outperforms GSV-Self and ImageNet-Self in recall values, underscoring its suitability for complex urban environments in VPR tasks. The GSV-Temporal model also excels on the Pitts30k dataset, achieving a recall value of 90.23% at K=15. This further emphasizes its capability in recognizing rapidly changing scenes. For the SPED dataset, GSV-Temporal displays superior recall values compared to other models, particularly with a notable performance at K=5, demonstrating its adaptability in diverse environments. Finally, in the MapillarySLS dataset, GSV-Temporal showcases its outstanding performance again, with a recall value of 77.57% at K=15, reinforcing its advantages in handling real-world scenarios.

In summary, the GSV-Temporal model consistently outperforms other models across multiple datasets, particularly in VPR tasks. Its sensitivity to temporal and environmental changes positions it as a superior choice for this application, revealing significant potential for practical use.

## 4.3 SOCIOECONOMIC INDICATOR PREDICTION

The socioeconomic indicator prediction task aims to use street view images to infer the socioeconomic status of urban areas. It emphasizes learning the macro atmosphere of a region rather than specific geometric features, highlighting the need for feature extraction to focus on similarities between regions to better understand economic conditions and developmental dynamics.

Table 1: Performance comparison on different datasets (Recall@K in %)

| Model | Dataset | k=1 | k=5 | k=10 | k=15 | k=20 | k=25 |
|---|---|---|---|---|---|---|---|
| ImageNet-Self | CrossSeason | 68.06 | 85.86 | 91.62 | 92.67 | 94.24 | 98.43 |
| GSV-Self | CrossSeason | 68.06 | 76.44 | 81.15 | 83.25 | 87.96 | 91.62 |
| GSV-Spatial | CrossSeason | 85.34 | 94.76 | 99.48 | 100.00 | 100.00 | 100.00 |
| GSV-Temporal | CrossSeason | **96.86** | **100.00** | **100.00** | **100.00** | **100.00** | **100.00** |
| ImageNet-Self | Essex | 62.38 | 84.29 | 90.00 | 95.24 | 96.67 | 98.57 |
| GSV-Self | Essex | 68.10 | 92.38 | 96.67 | 98.10 | 98.10 | 99.05 |
| GSV-Spatial | Essex | 76.19 | 92.38 | 97.14 | 98.10 | 98.57 | 98.57 |
| GSV-Temporal | Essex | **79.05** | **96.67** | **98.10** | **99.05** | **99.05** | **99.05** |
| ImageNet-Self | Pitts250k | 56.15 | 75.63 | 82.04 | 85.11 | 87.33 | 88.77 |
| GSV-Self | Pitts250k | 24.30 | 38.96 | 45.94 | 50.39 | 53.43 | 55.85 |
| GSV-Spatial | Pitts250k | 30.87 | 47.07 | 54.58 | 59.28 | 62.49 | 65.10 |
| GSV-Temporal | Pitts250k | **58.72** | **79.23** | **85.06** | **87.72** | **89.48** | **90.68** |
| ImageNet-Self | Pitts30k | 58.82 | 79.33 | 85.67 | 88.86 | 90.89 | 92.28 |
| GSV-Self | Pitts30k | 27.67 | 44.89 | 52.82 | 58.14 | 62.34 | 65.58 |
| GSV-Spatial | Pitts30k | 34.52 | 52.71 | 61.90 | 68.08 | 72.95 | 76.75 |
| GSV-Temporal | pitts30k | **64.11** | **82.26** | **87.51** | **90.23** | **91.58** | **92.65** |
| ImageNet-Self | SPED | 44.65 | 60.96 | 68.04 | 71.83 | 74.79 | 76.94 |
| GSV-Self | SPED | 36.24 | 51.73 | 57.50 | 60.30 | 63.92 | 67.22 |
| GSV-Spatial | SPED | 39.87 | 55.02 | 63.43 | 68.20 | 71.66 | 74.30 |
| GSV-Temporal | SPED | **50.08** | **66.06** | **72.82** | **75.78** | **77.27** | **79.90** |
| ImageNet-Self | MapillarySLS | 26.08 | 35.81 | 43.11 | 45.68 | 48.11 | 49.73 |
| GSV-Self | MapillarySLS | 20.27 | 29.86 | 34.59 | 37.16 | 38.92 | 41.22 |
| GSV-Spatial | MapillarySLS | 26.89 | 37.97 | 43.11 | 47.16 | 48.92 | 51.22 |
| GSV-Temporal | MapillarySLS | **54.19** | **69.32** | **75.27** | **77.57** | **79.86** | **81.62** |

In the downstream task of predicting socioeconomic indicators, we utilized the socioeconomic dataset published by Fan et al. (2023), which contains 18 socioeconomic indicators across seven major cities in the United States (Table A2). We take the socioeconomic indicator prediction of Los Angeles as an example. Detailed descriptions are provided in Section A.3. We first extracted street view embeddings from the images using the pre-trained models of local version. These embeddings were then aggregated at the block group level. The aggregated embeddings were used as input features to predict socioeconomic indicators for each block group.

For prediction model training and evaluation, we split each city's dataset into a training set (70%) and a testing set (30%). We used LASSO as the regressor to evaluate the predictive performance of the image embeddings extracted by the different pre-trained models. Additionally, we applied 5-fold cross-validation to ensure robust evaluation. This approach allows for a fair comparison of the different contrastive learning models in capturing visual features that are meaningfully correlated with socioeconomic indicators.

The results of socioeconomic indicator predictions are shown in Table 2. Overall, models pre-trained on street view images significantly outperform that pre-trained on the ImageNet dataset. Specifically, across all 18 indicators, the ImageNet-pretrained model achieved an average $R^2$ of 0.5209. In contrast, models on street view images achieved average $R^2$ scores of 0.5609 for self-contrastive, 0.5714 for temporal contrastive, and 0.5888 for spatial contrastive models, respectively. Furthermore, both temporal and spatial contrastive pre-training models capture more socioeconomic-related information compared to the self-contrastive approach, with spatial contrastive demonstrating the highest performance. This trend is consistent across most of socioeconomic indicators, showing the strongest predictive performance for Health-related indicators and the least for Crime-related indicators.

These findings suggest that spatial contrastive pre-training effectively captures the overall ambiance of urban areas, enabling more precise predictions of regional socioeconomic information. Additionally, temporal contrastive pre-training filters out random factors and dynamic elements in the images, enhancing the reliability of socioeconomic predictions.

Table 2: Performances of socioeconomic indicator prediction based on LASSO across models.

| Topic | Label | GSV-Self | GSV-Spatial | GSV-Temporal | ImageNet-Self |
|---|---|---|---|---|---|
| **Crime** | logcrime | 0.4203 | **0.4287** | 0.4194 | 0.4146 |
| | logpetty | 0.1810 | 0.1877 | **0.1892** | 0.1667 |
| | Total | 0.3007 | **0.3082** | 0.3043 | 0.2906 |
| **Health** | cancercrud | 0.6644 | **0.6969** | 0.6618 | 0.6053 |
| | diabetescr | 0.6589 | **0.6942** | 0.6796 | 0.6172 |
| | lpacrudepr | 0.8001 | **0.8337** | 0.8221 | 0.7671 |
| | mhlthcrude | 0.7088 | **0.7510** | 0.7291 | 0.6753 |
| | obesitycru | 0.7628 | **0.7886** | 0.7797 | 0.7175 |
| | phlthcrude | 0.7120 | **0.7399** | 0.7314 | 0.6752 |
| | Total | 0.7178 | **0.7507** | 0.7340 | 0.6763 |
| **Poverty** | mhincome_cbg | 0.6561 | **0.6816** | 0.6735 | 0.6096 |
| | povertyline_below100 | 0.1948 | **0.2227** | 0.1833 | 0.1718 |
| | povertyline_below200 | 0.6154 | 0.6377 | **0.6401** | 0.5893 |
| | Total | 0.4888 | **0.5140** | 0.4990 | 0.4569 |
| **Transport** | drove_alone_per_cbg | 0.3841 | **0.3991** | 0.3835 | 0.3582 |
| | estpmiles | 0.6196 | **0.6447** | 0.6289 | 0.5379 |
| | estptrp | 0.6024 | **0.6385** | 0.6087 | 0.5302 |
| | estvmiles | 0.6647 | **0.6921** | 0.6874 | 0.6163 |
| | estvtrp | 0.6900 | **0.6994** | 0.6991 | 0.6436 |
| | publictrans_per_cbg | 0.5226 | **0.5700** | 0.5339 | 0.4726 |
| | walkbike_per_cbg | 0.2383 | **0.2925** | 0.2340 | 0.2080 |
| | Total | 0.5317 | **0.5623** | 0.5394 | 0.4810 |
| **Overall Total** | | 0.5609 | **0.5888** | 0.5714 | 0.5209 |

## 4.4 SAFETY PERCEPTION

The safety perception task involves using street view imagery to estimate how safe people perceive a given scene to be. To make accurate estimates, this task requires analyzing all relevant elements within the scene, as each can contribute to the overall perception of safety, particularly elements such as trees and vehicles (Zhang et al., 2018).

Table 3: Evaluation Metrics of Different Models for Safety Perception Classification.

| Model | Accuracy (%) | Recall (%) | F1 Score (%) | AUC Score (%) |
|---|---|---|---|---|
| ImageNet-Self | 83.25 | 70.32 | 75.43 | 80.51 |
| GSV-Temporal | 84.91 | 65.16 | 75.94 | 80.72 |
| GSV-Spatial | 86.08 | 68.39 | 78.23 | 82.33 |
| GSV-Self | **88.68** | **77.42** | **83.33** | **86.29** |

We selected the PlacePlus 2.0 (Dubey et al., 2016) dataset for the downstream task of human environmental perception, filtering out over 1,144 images with safety perception scores below 3.5 and above 6.5, with 80% of the data used for training and 20% for testing. The model was trained using a linear binary classification approach for 20 epochs to effectively distinguish between low and high safety perception environments.

The evaluation metrics in Table 3 illustrate the performance of various models in classifying safety perception in urban environments. Notably, the GSV-Self model achieved the highest accuracy (88.68%) and recall (77.42%), demonstrating its effectiveness in identifying both safe and unsafe environments while minimizing false negatives. Its F1 score of 83.33% indicates a strong balance between precision and recall, and the AUC score of 86.29% further confirms its ability to distinguish between safety levels across thresholds. Overall, the GSV-Self model outperforms the others in all metrics, underscoring its potential for applications in urban safety perception tasks.

## 4.5 WHAT GSV-TEMPORAL AND GSV-SPATIAL CONTRASTS LEARN IN GSV?

Our experimental results reveal that different contrastive learning methods excel in different tasks: Temporal contrastive performs exceptionally well in VPR tasks, Spatial contrastive shows better results in macroeconomic prediction tasks, and Self contrastive achieves the best performance in

safety perception tasks, confirming our hypothesis. To further investigate the differences in model performance across these contrastive methods, we first visualized the attention mechanism in ViT and evaluated the attention range using attention distance.

**GSV-Temporal learns invariant characteristics, and GSV-spatial learns invariant neighborhood ambiance.**

This attention map visualization (Figure 2) shows how different contrastive learning strategies encode spatial and temporal invariants within urban street view images. The attention maps (Chefer et al., 2021) highlight how the models focus on distinct regions across various depths. We selected two street view images of the same location taken at different times. The attention maps for two query tokens, marked in red on the images, were visualized across layers from the first to the last depth, and the detailed results are available in Section A.5.

In the first depth, GSV-Self and GSV-Temporal exhibit a broader distribution of attention, while GSV-Spatial focuses more on localized regions. This suggests that GSV-Self and GSV-Temporal prioritize capturing global information in the early stages, whereas GSV-Spatial tends to emphasize detailed information initially. However, in the last depth, GSV-Self (Figure 2a, d) attends to global information across the image but tends to focus more on regions near the query token. In contrast, the GSV-Temporal model (Figure 2b, e) shows that query 1 (placed in the sky) primarily attends to the sky, filtering out dynamic elements. Query 2, placed on a car (a dynamic object), shows no attention to the car, reinforcing the model's ability to learn temporal-invariant characteristics by ignoring dynamic elements. In the GSV-Spatial model (Figure 2c, f), both query 1 and query 2 show similar attention patterns across the images. The model focuses on the overall structure without emphasizing dynamic objects like cars, indicating that spatial contrastive learning effectively captures spatial-invariant environmental characteristics. This supports the hypothesis that spatial contrast learning emphasizes the broader environment rather than individual objects.

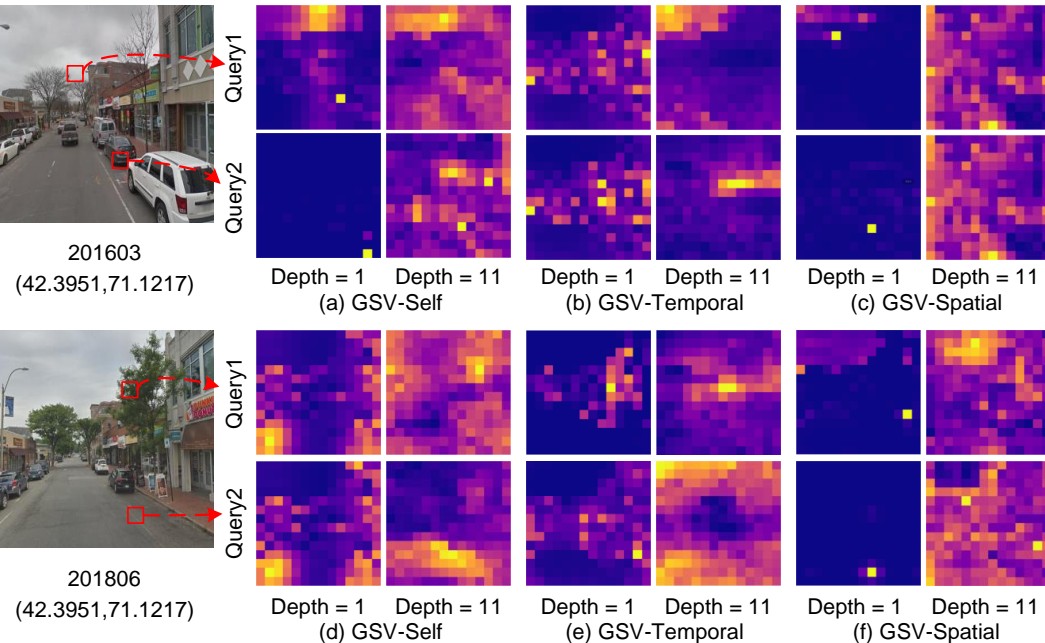

Figure 2: Attention maps for two queries visualized across models and depths. Red boxes indicate regions of focus. ImageNet-Self (a, d) emphasizes objects like cars. GSV-Temporal (b, e) filters out dynamic objects, highlighting static elements. GSV-Spatial (c, f) shows consistent focus across queries, capturing overall spatial structures.

We assess the spatial extent of self-attention by calculating attention distance (Dosovitskiy et al., 2021), to evaluate how different contrastive strategies focus on various aspects of the scene. Attention distance represents the mean distance between query tokens and key tokens, weighted by their respective self-attention scores. The figures illustrate the attention distances computed from sam-

pled street view images and ImageNet data. Specifically, GSV-Spatial exhibits the largest attention distance, indicating a tendency to focus on a broader spatial context rather than concentrating on individual objects. In contrast, the attention distances of GSV-Temporal and GSV-Self decrease sequentially, suggesting a gradual narrowing of focus to capture more specific details within the scenes. Notably, ImageNet-Self demonstrates the smallest attention distance, reflecting its pre-training on a dataset primarily consisting of object-centric images, which leads to a greater emphasis on individual objects over the overall spatial arrangement.

**GSV-Temporal exploits low-frequencies, and GSV-spatial exploits high-frequencies.**

we hypothesize that GSV-Temporal focuses on low-frequency information in scenes, while GSV-Spatial emphasizes high-frequency information. Specifically, in street view images, temporal-invariant characteristics of the built environment (such as static features like buildings and roads) typically exhibit lower frequencies, as these features remain stable over time with minimal variation. In contrast, spatial-invariant neighborhood ambiance (such as environmental features like lighting and weather) displays higher frequencies due to significant temporal and spatial variations. To validate this hypothesis, we report the relative log amplitude of Fourier-transformed representations by calculating the amplitude difference between the highest and lowest frequencies. Figure 3c and 3d illustrate the relative amplitude results for different contrastive learning strategies on street view images and ImageNet data.

The results show that the relative amplitude of GSV-Spatial is significantly greater than that of GSV-Temporal, indicating a stronger emphasis on high-frequency information in GSV-Spatial. Additionally, the model trained on ImageNet exhibits a greater focus on low-frequency features compared to street view images. These findings align with our hypothesis, further validating that GSV-Spatial effectively captures high-frequency details, while GSV-Temporal concentrates more on the low-frequency, stable aspects of the scene.

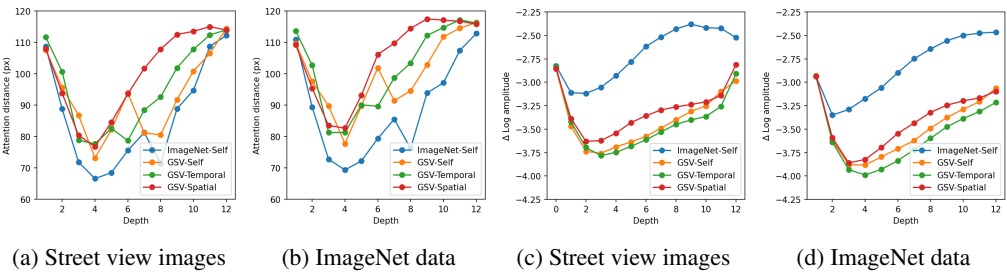

    (a) Street view images      (b) ImageNet data     (c) Street view images     (d) ImageNet data

Figure 3: Attention distance (a, b) and relative log amplitude (c, d) for different contrastive learning strategies on street view images and ImageNet data.

## 5 CONCLUSION

In conclusion, we propose a self-supervised urban visual representation framework based on street view images, capable of selectively extracting and encoding dynamic and static information and their ambiance in urban environments according to the requirements of different downstream tasks. By leveraging the unique spatiotemporal attributes of street view imagery, we have developed three contrastive learning strategies: temporal invariance representation, spatial invariance representation, and global information representation. Experimental results demonstrate that these strategies can effectively learn task-specific features suitable for their respective downstream applications, significantly enhancing performance in urban environment understanding tasks. Furthermore, we conducted an in-depth analysis of the reasons behind the performance of different contrastive methods, further emphasizing the importance of targeted learning strategies. This study systematically explores representation learning strategies based on street view images, provides a valuable benchmark for the application of visual data in urban science, and enhances their practical applicability.

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

# A APPENDIX

## A.1 STREET VIEW DATASET COLLECTION

To obtain street view imagery for both self-supervised model training and socioeconomic indicator prediction, we first sourced road network data for each city using the OSMnx library (Boeing, 2017) from OpenStreetMap. We then generated query points along these road networks at regular intervals of 15 meters. The Google Street View (GSV) Application Programming Interface (API) was subsequently utilized to retrieve and download street view images.

## A.2 PRE-TRAINING DATASET

As shown in Figure A1, the ten global cities include Amsterdam, Barcelona, Boston–Cambridge–Medford–Newton (Boston), Buenos Aires, Dubai–Sharjah (Dubai), Johannesburg, Los Angeles, Melbourne, Seoul, and Singapore. The details of street view datasets are presented in Table A1.

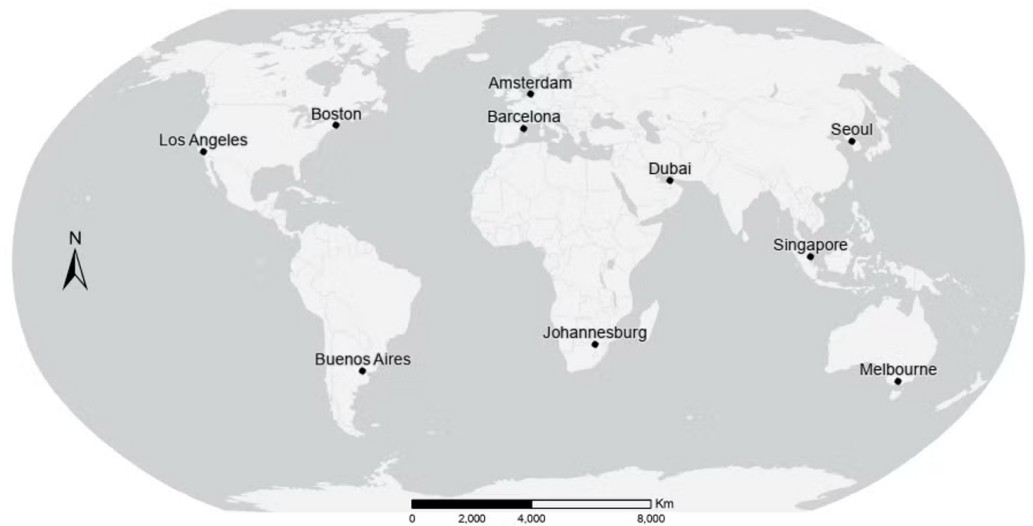

Figure A1: Spatial distribution of selected global cities.

## A.3 SOCIOECONOMIC INDICATOR PREDICTION DATASET

In our downstream task, we used socioeconomic indicators provided by Fan et al. (2023), which include data from seven major metropolitan areas in the United States. We take Los Angeles as an example. The socioeconomic indicators cover various topics relevant to urban studies and are detailed in Table A2.

Table A1: street view datasets for pre-training

| City | Country | # of images |
|------|---------|-------------|
| Amsterdam | Netherlands | 488,956 |
| Barcelona | Spain | 3,534,692 |
| Boston | United States | 9,295,736 |
| Buenos Aires | Argentina | 2,665,976 |
| Dubai | United Arab Emirates | 1,401,064 |
| Johannesburg | South Africa | 2,188,628 |
| Los Angeles | United States | 4,598,580 |
| Melbourne | Australia | 12,861,948 |
| Seoul | South Korea | 1,416,544 |
| Singapore | Singapore | 3,815,968 |

Table A2: Socioeconomic Indicators

| Topic | Indicator |
|-------|-----------|
| Poverty | Median Household Income |
| | % Individuals with poverty status determined: below 100% poverty line |
| | % Individuals with poverty status determined: below 200% poverty line |
| Health | Model-based estimate for crude prevalence of diagnosed diabetes among adults aged $\geq 18$ years |
| | Model-based estimate for crude prevalence of no leisure-time physical activity among adults aged $\geq 18$ years |
| | Model-based estimate for crude prevalence of obesity among adults aged $\geq 18$ years |
| | Model-based estimate for crude prevalence of cancer (excluding skin cancer) among adults aged $\geq 18$ years |
| | Model-based estimate for crude prevalence of physical health not good for $\geq 14$ days among adults aged $\geq 18$ years |
| | Model-based estimate for crude prevalence of mental health not good for $\geq 14$ days among adults aged $\geq 18$ years |
| Crime | Violent crime occurrence per spatial unit |
| | Violent theft-related crime occurrence per spatial unit |
| Transportation | % Population ($>16$) commute by walking and biking |
| | % Population ($>16$) commute by public transit |
| | % Population ($>16$) commute by driving alone |
| | Estimated vehicle miles traveled on a working weekday |
| | Estimated personal miles traveled on a working weekday |
| | Estimated vehicle trips traveled on a working weekday |
| | Estimated personal trips traveled on a working weekday |

## A.4 VISUAL PLACE RECOGNITION DATASET

**ESSEX.** The ESSEX dataset provides a diverse set of urban and suburban scenes with varying viewpoints and lighting conditions. It challenges the model's robustness in recognizing places despite changes in perspective and environmental factors (Zaffar et al., 2021).

**CrossSeason:** This dataset contains images captured across different seasons, aiming to study the impact of seasonal variations on image features. It is primarily used to train and evaluate models for visual recognition under varying seasonal conditions (Mans Larsson et al., 2019).

**Pittsburgh:** This is a large-scale dataset featuring street view images from Essex in the UK and Pittsburgh in the USA. It is designed to support visual localization and geographic scene recognition

tasks, providing rich environmental diversity suitable for various urban analysis studies (Arand-jelović et al., 2018).

**SPED:** This dataset focuses on the temporal changes in street view imagery, containing images of the same location captured at different time points. It aims to study the dynamic features of urban environments, suitable for temporal analysis and scene change detection (Chen et al., 2018).

**MapillarySLS:** This dataset includes street view images from around the globe, designed to support tasks in autonomous driving and visual understanding. Generated by users, it covers a variety of environments and conditions, providing rich geographical and scene information (Warburg et al., 2020).

A.5 ATTENTION

We visualized the attention maps across all depths for each contrastive learning strategy. The rows correspond to different strategies—ImageNet-Self, GSV-Self, GSV-Temporal, and GSV-Spatial—while the columns represent different depths (0 to 11). The original street view inputs are displayed on the left. Each attention map highlights the regions of the image that the model focuses on, demonstrating how attention shifts across depths for self, temporal and spatial features.

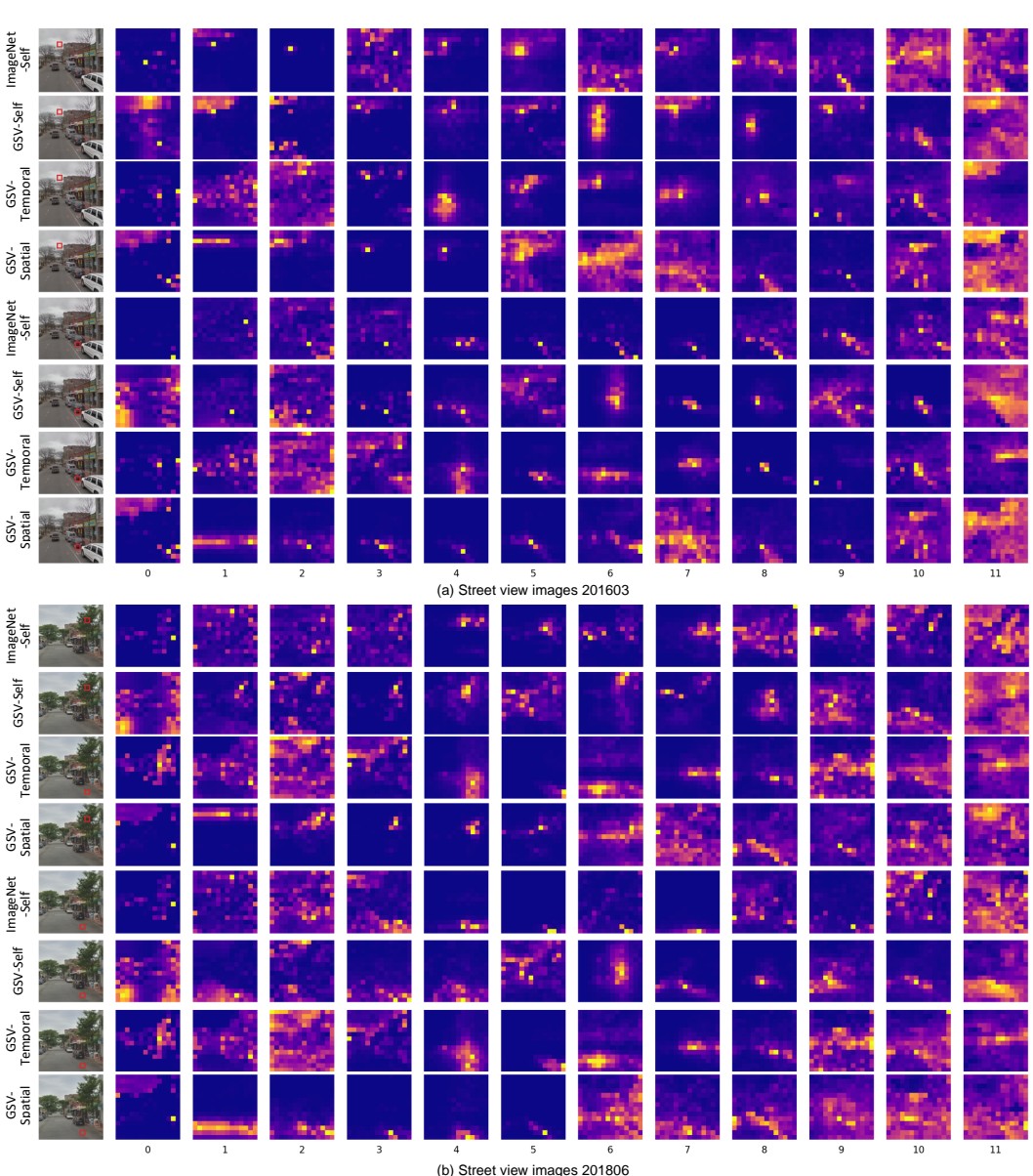

Figure A2: Attention maps for two queries visualized across models and depths.

