# OpenReview forum: "Spatiotemporal Contrast Are Natural Urban Scene Learners"
_ICLR.cc/2025/Conference — ICLR 2025 Conference Withdrawn Submission_

### Official Review · Reviewer_kmGn · 2024-10-18

**Soundness:** 2
**Presentation:** 3
**Contribution:** 3
**Rating:** 5
**Confidence:** 4

**Summary:**

This paper introduces a self-supervised learning framework designed to extract both dynamic and static information from street view images for various urban environment tasks. It presents three contrastive learning strategies—temporal invariance, spatial invariance, and global information representation—that leverage the spatiotemporal characteristics of street view imagery. These strategies are tailored to different downstream tasks such as visual place recognition, socioeconomic prediction, and safety perception. The experimental results show that these approaches significantly enhance performance in task-specific urban analyses. Moreover, the paper provides a detailed exploration of how each contrastive learning method captures features relevant to its respective task, offering a valuable benchmark for urban science applications.

**Strengths:**

1. **Strong Innovation**:
   The paper introduces a self-supervised learning framework based on street view imagery that effectively extracts and encodes both dynamic and static information from urban environments. The framework selectively adapts to different downstream tasks. This approach demonstrates strong innovation, particularly in leveraging spatiotemporal attributes to capture urban environment invariance and neighborhood ambiance, which is a notable contribution to urban environment research.

2. **Comprehensive Contrastive Learning Strategies**:
   The authors design three distinct contrastive learning strategies—temporal invariance representation, spatial invariance representation, and global information representation. Experimental results confirm that these strategies perform well across different tasks. This demonstrates the effectiveness of task-specific learning, enhancing model performance in tasks such as visual place recognition, socioeconomic prediction, and safety perception.

3. **Extensive and Convincing Experiments**:
   The experimental section covers a broad range of downstream tasks and evaluates performance on several datasets. The results clearly show the effectiveness of the proposed framework and strategies across various tasks. By utilizing multiple benchmark datasets (such as CrossSeason and Pitts250k), the experimental findings are both comprehensive and representative.

4. **In-depth Model Interpretability Analysis**:
   The paper not only presents experimental results but also provides analysis through attention map visualization and frequency analysis, helping readers understand how different contrastive learning strategies capture temporal and spatial invariants. This enhances the model’s interpretability, offering insights into how it captures global and temporal information.

**Weaknesses:**

1. **Lack of Methodological Details**:
   The method section of the paper is too brief, particularly in describing key elements such as data augmentation, positive and negative sample pair construction, contrastive learning training details, the choice of backbone, and whether fine-tuning was performed. It is recommended to include more details about these aspects, such as the data augmentation techniques, the ratio of positive and negative samples, training hyperparameters (e.g., learning rate, batch size), and the strategy for fine-tuning the backbone. This will improve the reproducibility and clarity of the methodology.

2. **Insufficient Experimental Comparison**:
   The current experiments only compare the proposed approach to ImageNet-Self, lacking comparisons with mainstream methods in the field, such as Urban2Vec. Since this paper introduces a new contrastive learning strategy, it would be beneficial to apply this strategy to existing models and evaluate its performance in different frameworks. Additionally, more comparison experiments with other mainstream models would further verify the applicability and advantages of the proposed method.

3. **Absence of Ablation Studies**:
   The main innovation of this paper appears to lie in the data construction rather than the model design, making it necessary to explore how the data construction impacts the model's performance. An ablation study is recommended to analyze the effects of modifying aspects such as the time window, spatial range, sample pair construction rules, and data augmentation strategies. This would help to clarify the critical role that data construction plays in determining the effectiveness of the model.

**Questions:**

1. **Methodological Details**:
   - Could you clarify the specific data augmentation techniques used in the training process? How do these augmentations affect the formation of positive and negative sample pairs, and how are the positive/negative ratios chosen?
   - Can you provide more details about the choice of the backbone (e.g., Vision Transformer)? Was any part of the backbone fine-tuned during the training process, or was it entirely frozen? If fine-tuned, which layers were adapted, and why?

2. **Comparison with Mainstream Baselines**:
   - The experiments compare with ImageNet-Self, but other methods like Urban2Vec or similar spatial-temporal learning approaches are not included. Have you considered integrating your contrastive strategy into these existing methods to verify whether the proposed approach can enhance their performance? If not, what are the main challenges in doing so?

3. **Ablation Studies**:
   - Since much of the innovation seems to focus on the data construction, it would be helpful to understand the impact of different data construction strategies on the model's performance. Have you considered conducting ablation studies by modifying factors such as time window size, spatial region definition, or the construction of positive and negative pairs? How do you expect these factors to affect the results?

---

### Official Review · Reviewer_n1Bm · 2024-10-29

**Soundness:** 3
**Presentation:** 2
**Contribution:** 2
**Rating:** 5
**Confidence:** 4

**Summary:**

This work explores the temporal and spatial attributes of street view images and proposes an unsupervised learning framework suitable for various downstream tasks. Three downstream tasks were designed to validate the model's effectiveness across multiple cities, and relevant visualization analyses were conducted.

**Strengths:**

1. A dataset of 42 million street view images across 10 cities, with extensive coverage. If released as open source, it would greatly benefit the community.

2. The framework is designed to be simple and easily extendable to various downstream tasks.

**Weaknesses:**

1. This work merely applies MoCo to the spatial and temporal dimensions, lacking innovation.

2. The dataset used for the safety perception task is still the one introduced in a 2016 paper, which raises concerns about its timeliness.

3. Since the framework includes temporal and spatial branches, why not design a fusion module to comprehensively leverage spatiotemporal features? This gives the impression that your model design does not fully utilize the data.

4. In the experiments, comparisons with previous baselines should not only include works using different datasets but also different model architectures on the same dataset, which is lacking here. For example, methods mentioned in the related work, such as CPC, SeCo, and Geo-SSL, are all relevant for comparison.

**Questions:**

Please see weakness.

---

### Official Review · Reviewer_XAbY · 2024-10-29

**Soundness:** 2
**Presentation:** 3
**Contribution:** 2
**Rating:** 3
**Confidence:** 4

**Summary:**

The paper proposes a contrastive learning-based self-supervised learning to learn different sets of feature representations for street view images. The key idea is to build 3 different kinds of contrastive representations with different pretext tasks namely: self-supervision (instance classification), temporal contrast (images from the same location/angle are similar), and spatial contrast (images from nearby locations are similar). The paper optimizes the MoCo framework to optimize a vision transformer for these three pretext tasks. The results suggest that different representations work better for different downstream tasks e.g. for visual place recognition, the temporal model works best on the other hand for socioeconomic indicator prediction the spatial model works best. The paper also does some analysis of feature representation to show why a specific representation might be better for a specific set of downstream tasks.

**Strengths:**

* The paper justifies the model's performance on a comprehensive set of downstream tasks, including 5 different datasets for visual place recognition and 18 different socioeconomic indicators.

* The paper is well-motivated, and the method is well-written and easy to understand. The problem of understanding which features are useful for a specific downstream is crucial to understand in many areas and this work does a good job of it targeted on streetview image.

**Weaknesses:**

The method lacks novelty:
   * Spatio-temporal contrastive learning has been very well explored and implemented in a lot of domains including videos and satellite images [SeCo for example]. As a result, the self-supervised method proposed paper lacks novelty.
   * Similar self-supervised methods have also been well-explored in the domain of street view (or location-to-image pairs), with [1] being very similar and [2] being slightly related.
   * The paper does not go beyond these well-explored ideas of spatio-temporal contrast and therefore lacks technical novelty.

Lack of baselines/bounds:
   * The paper does not compare to any baselines other than the models proposed in the paper itself. As a result, it is hard to justify if the paper answers the main question of the paper "Can spatiotemporal contrast learn Natural Urban Scenes?" It is unclear is the numbers reported in Table 1,2,3 are good because there is no baseline to compare against other than the Imagenet, which is very object-centric and can be treated as a different domain.
   * More baselines such as models trained on places datasets, and more scene-centric datasets would be useful to evaluate.

   * On similar lines, a fully-supervised oracle model can also be used, to understand how close the self-supervised methods are to some "upper bound".

   * Foundation model research in vision has also explored masked image modeling as an alternative to contrastive learning. It would be interesting to see how well spatio-temporal contrast fares against these and other self-supervised baselines.

   * Foundation model research also typically envisions a single model that can produce rich features for a diversity of tasks. Combinations of these list of features can also be tested, for example how well a GSV-spatial+GSV-temporal model performs on the task.

   * Since a total of 42 million images are collected, it might also be useful to see the scaling law of this method. At that scale does it even matter to do spatio-temporal contrast? The SeCo paper indicated that having inductive biases such as spatio-temporal contrast is more useful when the number of images is fewer. The model can also be compared against imagenet-21k based self-supervised baselines.


Paragraph 1 of the introduction proposes several statements as natural hypotheses for example: **human perception of places require dynamic elements** and similarly for other tasks. These hypotheses are not very clear to me and would not be the first thing that naturally comes to my mind. For example, why would the human perception of safety require dynamic elements however the actual safety (or crime: a socio-economic indicator) does well with spatially consistent expressions. Either some citations should be provided or this should not be posed as hypotheses to be proved later in the paper with data.

The qualitative results are hard to understand.
* I don't see many arguments made in the paper for the qualitative results in the figure. For example, "GSV-Self and GSV-Temporal prioritize capturing global information in the early stages, whereas GSV-Spatial tends to emphasize detailed information initially". I don't see this happening in the figures, for example in A2, in many cases, GSV-self is also emphasizing detailed information initially (the first GSV self-row). The other statements in that paragraph are also similarly hard to justify from the figure.

* Regarding the frequency plot: the amplitudes are measured in Fourier transform in the feature space. Since the feature transformation (with a ViT) is very non-linear, high frequency in the feature space does not necessarily imply high frequency in the input space. Thus it is unclear if the conclusion "GSV-Temporal exploits low-frequencies" can be made.

* Is an attention map or frequency the best way to understand the feature space? Alternatives could include feature distance/retrieval on unseen streetview images. For example, can GSV-temporal retrieve the same scene even if the dynamic objects in the scene change. What do the other models retrieve?



[1] Stalder, Steven, Michele Volpi, Nicolas Büttner, Stephen Law, Kenneth Harttgen, and Esra Suel. "Self-supervised learning unveils change in urban housing from street-level images." arXiv preprint arXiv:2309.11354 (2023).

[2] Vivanco Cepeda, Vicente, Gaurav Kumar Nayak, and Mubarak Shah. "Geoclip: Clip-inspired alignment between locations and images for effective worldwide geo-localization." Advances in Neural Information Processing Systems 36 (2024).

**Questions:**

Some clarification questions/suggestions:
At lines 359-360: when the dataset is split into training and testing, can images from the same block group belong to both training and test set? The reason for asking this question is that this decision can have a huge impact on performance. If the answer is yes, then the spatial model's ability to learn (memorize) block group (that is being done at training time) can help a lot. Whereas, if the answer is no, the spatial model is indeed generalizing. This should be made clear in the paper.

LASSO regression should be cited.

Line 250 has a type: We *collected* historical images

Table 1 has a bit of redundancy, as it shows just the recall values at different Ks, it can be replaced by recall curves.

More details on data collection should be provided.

---

### Official Review · Reviewer_fc71 · 2024-11-04

**Soundness:** 2
**Presentation:** 2
**Contribution:** 2
**Rating:** 3
**Confidence:** 4

**Summary:**

This study presents a self-supervised urban visual representation framework based on street view images, which selectively extracts and encodes dynamic and static information, along with their ambiance, according to the needs of various downstream tasks. By leveraging the unique spatiotemporal attributes of street view imagery, three contrastive learning strategies are developed: temporal invariance representation, spatial invariance representation, and global information representation. Experimental results demonstrate that these strategies effectively learn task-specific features, significantly enhancing performance in urban environment understanding tasks. Additionally, an in-depth analysis of the performance reasons behind different contrastive methods emphasizes the importance of targeted learning strategies.

**Strengths:**

The originality of this paper lies in its use of different types of contrastive learning strategies tailored to various tasks involving street view images. This approach not only considers the diversity of tasks but also flexibly extracts and encodes dynamic and static information from street view imagery through strategies such as temporal invariance representation, spatial invariance representation, and global information representation.

The paper is written very clearly.

**Weaknesses:**

The methods presented in this paper lack sufficient innovation. This appears to be merely an incremental innovation, essentially applying MoCo along with A/B/C methods. The three hypotheses proposed by the authors have already been widely applied in the fields of satellite imagery and even natural imagery.

[1] K. Ayush et al., "Geography-Aware Self-Supervised Learning," 2021 IEEE/CVF International Conference on Computer Vision (ICCV), Montreal, QC, Canada, 2021, pp. 10161-10170, doi: 10.1109/ICCV48922.2021.01002.

[2] Wang, Z., Li, H. and Rajagopal, R. 2020. Urban2Vec: Incorporating Street View Imagery and POIs for Multi-Modal Urban Neighborhood Embedding. Proceedings of the AAAI Conference on Artificial Intelligence. 34, 01 (Apr. 2020), 1013-1020. DOI:https://doi.org/10.1609/aaai.v34i01.5450

[3] Yanxin Xi, Tong Li, Huandong Wang, Yong Li, Sasu Tarkoma, and Pan Hui. 2022. Beyond the First Law of Geography: Learning Representations of Satellite Imagery by Leveraging Point-of-Interests. In Proceedings of the ACM Web Conference 2022 (WWW '22). Association for Computing Machinery, New York, NY, USA, 3308–3316. https://doi.org/10.1145/3485447.3512149

**Questions:**

Q1: L41-46: I'm a bit confused by your examples. In localization tasks, dynamic information is also important, such as vehicle brand, sun direction, and types of greenery (is this static information?). The relevant features in tasks related to human perception of places are also not entirely accurate; static features are important as well.

Q2: L49-50: You've only listed annotation-related challenges here. What about the challenges during training? Could you provide more specific details on the annotation issues? It’s a bit confusing.

Q3: What is the purpose of Figures 3a and 3b, and how were they obtained? Could Figures 3c and 3d be expressed with equations to show how they were derived? How much data was used to create this figure?

Q4: Why didn't you compare other baselines for each task, such as UrbanCLIP, GeoCLIP, etc.?  Since you have utilized performance from such a wide range of public datasets, could you showcase the optimal performance of specific models corresponding to these datasets? This would help demonstrate the advantages of your contrastive learning approach. How do tasks like scene classification and object detection perform?

Q5: In urban understanding tasks, self-supervised learning is typically used to enable a single model to adapt to various downstream tasks. However, in your approach, each task corresponds to a separate model. Why not adopt an end-to-end approach instead?

Q6: In your task of socioeconomic indicator prediction metrics, your model performs well in the selected city dataset. However, how does it perform when transferred to other cities? I believe that demonstrating superior performance in the transfer study would strongly highlight the application value of your model.

Q7: Why is SAFETY PERCEPTION treated as a binary classification task? Does this task setting seem overly simplistic?

Q8:  Does your model use pre-trained parameters? If so, which ones are used? If it does, it seems to be incremental pre-training, and I'm curious about the performance of other self-supervised models on your dataset.

Q9: Will you be open-sourcing the dataset and code? Currently, it seems that your paper's main contribution is the dataset. If not, please state the primary contribution of your paper

---

### Note · Authors · 2024-11-26

I have read and agree with the venue's withdrawal policy on behalf of myself and my co-authors.